# Design of a Cost-Effective Ultrasound Force Sensor and Force Control System for Robotic Extra-Body Ultrasound Imaging

**DOI:** 10.3390/s25020468

**Published:** 2025-01-15

**Authors:** Yixuan Zheng, Hongyuan Ning, Eason Rangarajan, Aban Merali, Adam Geale, Lukas Lindenroth, Zhouyang Xu, Weizhao Wang, Philipp Kruse, Steven Morris, Liang Ye, Xinyi Fu, Kawal Rhode, Richard James Housden

**Affiliations:** 1School of Biomedical Engineering and Imaging Sciences, King’s College London, London SE1 7EH, UK; hongyuan.ning@kcl.ac.uk (H.N.); eason.rangarajan@kcl.ac.uk (E.R.); aban.merali@kcl.ac.uk (A.M.); adam.t.geale@kcl.ac.uk (A.G.); lukas.lindenroth@kcl.ac.uk (L.L.); zhouyang.xu@kcl.ac.uk (Z.X.); weizhao.wang@kcl.ac.uk (W.W.); philipp.kruse@kcl.ac.uk (P.K.); steven.c.morris@kcl.ac.uk (S.M.); kawal.rhode@kcl.ac.uk (K.R.); richard.housden@kcl.ac.uk (R.J.H.); 2School of Medicine, Southern University of Science and Technology, Shenzhen 518055, China; 12112644@mail.sustech.edu.cn (L.Y.); 12112743@mail.sustech.edu.cn (X.F.)

**Keywords:** robot-assisted ultrasound imaging, force control for robotic systems, force sensor design, medical robotics, multi-sensor fusion, ultrasound-compatible phantom design

## Abstract

Ultrasound imaging is widely valued for its safety, non-invasiveness, and real-time capabilities but is often limited by operator variability, affecting image quality and reproducibility. Robot-assisted ultrasound may provide a solution by delivering more consistent, precise, and faster scans, potentially reducing human error and healthcare costs. Effective force control is crucial in robotic ultrasound scanning to ensure consistent image quality and patient safety. However, existing robotic ultrasound systems rely heavily on expensive commercial force sensors or the integrated sensors of commercial robotic arms, limiting their accessibility. To address these challenges, we developed a cost-effective, lightweight, 3D-printed force sensor and a hybrid position–force control strategy tailored for robotic ultrasound scanning. The system integrates patient-to-robot registration, automated scanning path planning, and multi-sensor data fusion, allowing the robot to autonomously locate the patient, target the region of interest, and maintain optimal contact force during scanning. Validation was conducted using an ultrasound-compatible abdominal aortic aneurysm (AAA) phantom created from patient CT data and healthy volunteer testing. For the volunteer testing, during a 1-min scan, 65% of the forces were within the good image range. Both volunteers reported no discomfort or pain during the whole procedure. These results demonstrate the potential of the system to provide safe, precise, and autonomous robotic ultrasound imaging in real-world conditions.

## 1. Introduction

Ultrasound imaging, first introduced in the 1940s, has become a widely utilized diagnostic modality due to its radiation-free nature, portability, real-time imaging capabilities, and greater accessibility compared to computed tomography (CT) scans and magnetic resonance imaging (MRI) [1]. However, traditional ultrasonography depends heavily on skilled operators, resulting in variability in image quality and reproducibility [2]. Additionally, it poses risks to operators, such as musculoskeletal disorders and increased infection risk when scanning patients with infectious diseases [3]. Robot-assisted ultrasound imaging may addresses these challenges by enabling consistent, precise, and faster scans, reducing human error and lowering healthcare costs [4] while simultaneously mitigating the risk of operator injury and infection. A critical factor in advancing robotic ultrasound systems is the implementation of effective force control, which ensures optimal contact between the probe and the patient’s body. Excessive contact force can lead to patient injury, whereas insufficient force compromises image quality [5]. Researchers adopt diverse approaches to force control in ultrasound robots, involving varying choices of sensors, control strategies, and implementation methodologies.

Robotic ultrasound systems employ various types of force sensors to ensure precise and safe interactions between the ultrasound probe and the patient’s body. Among these, external commercial force/torque sensors are widely used due to their accuracy and reliability, making them a preferred choice for precise force measurements. For instance, many systems utilize six-axis force/torque sensors, typically positioned between the robot arm and the probe, as demonstrated in studies by Merouche et al., Jiang et al., Kim et al., and Chen et al. [6,7,8,9]. Alternatively, some systems adopt different configurations, such as the approach described by Huang et al., which employs two one-axis sensors mounted on the front face of the ultrasound probe to measure the contact force between the probe and the patient [10]. Beyond commercial sensors, custom-built sensors are also explored. Noh et al. developed a six-axis force/torque sensor using 3D-printing technology, demonstrating the feasibility of designing sensors tailored to specific tool shapes and functional requirements; however, this sensor has not yet been validated in clinical settings [11]. Additionally, certain robotic ultrasound systems utilize built-in sensors integrated into the robot’s structure, offering a compact and streamlined solution for force feedback. Examples of such systems are detailed in the works of Goel et al., Suligoj et al., Virga et al., and Hennersperger et al. [12,13,14,15]. Each type of sensor—whether commercial, custom-built, or built-in—plays a pivotal role in enhancing the performance and accuracy of robotic ultrasound systems, with the choice of sensor often guided by the specific application requirements and system constraints.

The force control strategies for maintaining appropriate contact force during robotic ultrasound procedures vary across systems, each offering distinct advantages. A common strategy is applying a constant force, as implemented in [6,9,13]. This approach provides simplicity and reliability, ensuring uniform probe–patient interaction that supports consistent imaging outcomes and basic patient comfort. Another approach involves maintaining contact force within a specific range, as seen in [10,15], where the force is dynamically adjusted to predefined limits. This strategy allows for a balance between safety and effectiveness by ensuring sufficient tissue contact to capture clear images while preventing excessive pressure that could lead to discomfort or injury. A more advanced method is applying a patient-specific contact force, utilized in [7,8,14]. These systems tailor the force based on real-time feedback, such as ultrasound image quality or confidence maps, optimizing diagnostic accuracy by adapting to individual patient anatomy and imaging needs. Each of these strategies reflects a trade-off between simplicity, adaptability, and personalization in robotic ultrasound applications.

Various force control methods are employed to regulate the interaction force between the ultrasound probe and the patient, each offering specific advantages. Position control, as implemented in [6], relies on precise robotic arm movements to maintain the desired force, providing a straightforward approach for force regulation. Impedance control, highlighted in [9,13], adjusts the robot’s compliance based on measured forces, enabling safer and more consistent contact by dynamically responding to force variations. Hybrid position–force control, used in [8,10,12], combines the strengths of both position and force control to dynamically modulate the probe’s position and applied force. This approach ensures optimal contact with the patient and enhances image quality by balancing movement precision with force regulation. Compliance control, described in [15], maintains a constant desired force within a specified range, allowing the system to adapt to varying tissue characteristics while ensuring both patient safety and imaging accuracy.

In addition to these force control strategies, some studies incorporate additional mechanical safety features to limit the maximum force, providing an extra layer of protection. Housden et al. [16] developed a spring-ball-based mechanical clutch to mechanically restrict the force to a safe limit. Similarly, Tsumura et al. [17] employed a constant-force spring mechanism that, combined with a counterweight system, ensures a steady and limited contact force by offsetting the probe’s weight and directly controlling the applied force. These mechanical safety features ensure that the interaction force remains within safe limits, enhancing patient safety during robotic ultrasound procedures.

In conclusion, current robotic ultrasound systems exhibit a diverse range of force sensors and control strategies. However, most of these systems rely on either standardized and costly commercial force sensors or the built-in sensors of commercial robot arms. There is a notable lack of solutions specifically designed for ultrasound scanning that utilize lightweight and cost-effective sensing systems for custom-made robots, which often lack sophisticated internal force sensors.

In this paper, we present a solution for automated extra-body ultrasound scanning, with a comprehensive robot control framework as the core contribution. Supporting this framework are innovations in sensor design, algorithm development, and validation through realistic testing, collectively advancing the feasibility of automated ultrasound in clinical applications. The key contributions of our work are as follows:(1)Robot Control Framework: The primary contribution of this work is the development of a comprehensive robot control framework for automated ultrasound scanning. This framework integrates patient-specific surface-based path planning, operator inputs, and force-control algorithms, ensuring safe and consistent acquisition of high-quality ultrasound images. This framework represents a complete and robust solution for automated scanning in real-world scenarios.(2)Force Sensor Design: To support the control framework, we designed a compact, single-axis force sensor using load cells that can be seamlessly integrated into the ultrasound probe. This sensor is optimized for 3D printing, reducing fabrication costs while providing reliable and accurate force measurements necessary for implementing the control algorithms.(3)Phantom Design: We developed a two-layer abdomen phantom, featuring a stiff inner layer with an abdominal aortic aneurysm (AAA) and inferior vena cava (IVC) and a soft adipose-mimicking outer layer. This phantom closely resembles human tissue, allowing realistic testing of force-control algorithms and ultrasound imaging in a simulated clinical setting.(4)System Validation: We validated the system’s performance through both phantom and healthy volunteer experiments, demonstrating its effectiveness in real-world scenarios.

This paper is organized into two main parts: sensor design and characterization, followed by robot control design and system performance testing. In the methodology section, Section 2.2 and Section 2.5.1 focus on the design and characterization of the custom force sensor, detailing its mechanical structure, signal processing, and calibration experiments. Section 2.3, Section 2.4, Section 2.5.2 and Section 2.5.3 describe the robot control algorithms, phantom design, and experimental setups for validating the system’s performance. In the results section, Section 3.1 presents the outcomes of the sensor characterization experiments, including calibration, accuracy, and performance under real-world conditions. Section 3.2 and Section 3.3 evaluate the robot control algorithms and system performance, showcasing the integration of force control, path planning, and automated ultrasound image acquisition through both phantom and volunteer experiments. Finally, Section 4 and Section 5 discuss and conclude this research, respectively.

## 2. Materials and Methods

The aim of this research is to develop a system for automated ultrasound scanning of a target region, ensuring consistent force application within a predefined safe range. To achieve this, we began by implementing system registration and calibration to ensure precise alignment and tracking between the patient and the robotic system. We then designed and fabricated a cost-effective, custom-built force sensor tailored for ultrasound applications. Building on this, we developed a force control strategy to maintain optimal probe contact during scanning, ensuring patient safety and high-quality imaging. Finally, we constructed a detailed, ultrasound-compatible phantom as a realistic testing platform to validate the system’s functionality before advancing to patient trials.

### 2.1. Robotic System Setup and Workflow

This research advances a dual-arm robotic system, originally developed by our research team at King’s College London, for enhanced prenatal ultrasound screenings as detailed in [18]. Featuring 17 degrees of freedom, the robot’s end effector is equipped with an ultrasound holder that mimics the dexterity of a human sonographer’s wrist. It is controlled through custom software developed in C++ using Qt Creator 8.0.2 (based on Qt 6.3.2, GCC 10.3.1, The Qt Company Ltd., Espoo, Finland). The software not only allows real-time visualization of the robot’s pose and the patient’s surface mesh but also supports various control methodologies, including motor and endpoint control. This makes it highly suitable for both research and clinical applications. Additionally, the system is ethically approved for abdominal scans, including use with pregnant women. The system incorporates patient-to-robot registration and real-time movement detection using two 3 cm × 3 cm AprilTag markers—one placed on the patient and the other on the robot—along with an Intel RealSense RGB-D camera, as shown in Figure 1. This setup enables the robot to continuously track the patient’s position and dynamically adjust for any patient movement. Additional details can be found in our previous work [19].

The general workflow of our current system for autonomous robotic ultrasound acquisitions is illustrated in Figure 2. The system operates in two main stages: Before Acquisition and During Acquisition. In the Before Acquisition stage, an external RGB-D camera captures the patient’s surface information, which enables real-time patient-to-robot registration using AprilTag markers. This process ensures that the scanned patient surface aligns correctly within the robot control’s simulation environment, allowing the sonographer to precisely select the desired scanning region in the software. The robot then moves to the corresponding location on the patient’s body to perform the scan. During the image acquisition stage, the robot autonomously follows a scanning path defined by the sonographer while actively monitoring and adjusting contact force via force sensors. It dynamically modifies the path in response to patient movement and force feedback, ensuring safe and accurate image acquisition. Upon completion, the acquired ultrasound images can be transferred to healthcare professionals or AI-based algorithms for further assessment, facilitating diagnostic support or advanced analysis.

### 2.2. Force Sensor Design for Ultrasound Probe Compatibility

#### 2.2.1. Mechanical Structure Design

In most robotic ultrasound systems, the force sensor is typically positioned either at the probe face to measure the direct force between the patient and the probe [10] or at the top of the probe [6,7,9], between the robot end and the probe, for ease of mounting on the robot. However, placing the sensor on the probe face limits the sensor size and can cause shadowing on the ultrasound image. On the other hand, mounting the sensor at the top of the probe leads to indirect and less accurate force measurements due to the complexity of force transmission.

To overcome these challenges, our goal is to design a round-shaped force sensor that holds the probe similarly to a human sonographer, inspired by and building upon our previous model [11]. We aim to create a sensor that is not only affordable and easy to manufacture but also widely accessible to researchers and industries around the world. By focusing on a straightforward structure and force-sensing principle, we can ensure that the sensor is easy to repair if needed. Additionally, since this sensor is intended for clinical use, it is designed to be robust and capable of long-term operation without failure. Lastly, the sensor is designed to be adaptable to a wide range of commonly used ultrasound probe types and robotic systems, ensuring broad applicability for various research and clinical applications.

To meet these requirements, we have selected the FX29 Compact Compression Load Cell from TE Connectivity as the basic element. These load cells are cost-effective, readily available, and can be easily replaced if damaged, avoiding the need to rebuild the entire sensor. The proposed sensor consists of four parts, as is shown in Figure 3. The left and right parts are separate to fit the probe, while the upper and lower parts are separate to facilitate easy replacement of the load cells. The lower part is designed to hold the probe firmly, with its inner shape derived from the probe’s CT scan. The upper part is designed to fit the sensor into the robot’s end effector based on the robot’s CAD model. The left-right and upper-lower parts of the sensor can be easily modified and reprinted to accommodate different probe shapes and robotic configurations, making the sensor highly adaptable.

The load cells are evenly distributed on the lower plate, providing a stable structure. Eight cylinders extend from the top half; four are designed to push the load cells at their centres, with lengths matching the upper and lower gap to prevent undue pressure on the load cells when unloaded. The remaining four cylinders are supporting cylinders with tiny bulges at their ends (Figure 3d). By applying pressure, these supporting cylinders fit securely into corresponding holes in the bottom plate, ensuring that upper and lower parts fit together during scans. The holes for the supporting cylinders are designed to be 2 mm deeper than the cylinders themselves. This 2 mm gap exceeds the maximum deformation expected in the load cells, allowing the supporting cylinders to move freely downward under pressure without transferring force. The entire sensor structure is fabricated using 3D-printed PLA material, optimizing for both function and manufacturing efficiency.

The sensor is calibrated as a single-axis force sensor, meaning it only measures forces perpendicular to the plane of the four load cells. The total force ($F_{z}$) is computed by summing the readings from the four load cells. This calibration ensures accurate measurement of vertical forces, which is used in subsequent force-control algorithms.

#### 2.2.2. Force Sensor Signal Processing

In this research, we implemented a load cell system using four load cells with I2C communication, each with four wires: power, ground, clock, and data. The load cells were assigned unique I2C addresses and connected to a shared I2C bus using ribbon cables for improved organization and reliability. The system features a custom-made printed circuit board (PCB) equipped with pull-up resistors and a zero (tare) button, which is crucial for calibrating the load cells before operation (Figure 4b). The I2C signals are processed by a Teensy 4.1 microcontroller (PJRC, Sherwood, Oregon, USA) housed in a custom 3D-printed enclosure mounted beneath the robot arm (Figure 4c). The Teensy 4.1 reads the load cell data from the PCB and transmits them to the robot’s main microcontroller for force control. This setup not only ensures seamless integration of load cell data with the robot’s control system but also allows for easy replacement of damaged load cells and direct signal access for debugging purposes (Figure 4d).

### 2.3. Robot Control Algorithm Design

#### 2.3.1. Force Control Strategy Design

To ensure safe and effective scanning, we implemented a force control strategy that dynamically adjusts the probe’s position to maintain the contact force within a patient-specific, predefined range. This range is designed to balance safety and image quality during extra-body ultrasound procedures. The “force” referred to here is ($F_{z}$), the total vertical force calculated as the sum of the four load cells, as described in Section 2.2.1.

Our system employs a proportional (P) control strategy, a simple yet effective approach for maintaining consistent force. The system continuously monitors the contact force and calculates the deviation from the desired range. If the current force is within the range, no adjustment is needed. However, if the force exceeds or falls below the desired range, an adjustment is applied to the probe’s position to correct the error.

The adjustment distance, $d_{\mathrm{adjusted}}$, is calculated as:$$d_{\mathrm{adjusted}}=d_{\mathrm{current}}+k_{p}\cdot F_{\mathrm{error}}$$
where $F_{\mathrm{error}}$ represents the deviation from the target force range, and $k_{p}$ is the proportional control coefficient tuned based on the softness of the tissue. Softer tissues require higher $k_{p}$ values for greater sensitivity to force changes, while firmer tissues require lower $k_{p}$ values to prevent overcorrections.

To enhance the system’s precision, the robot calculates the normal direction of the surface at each contact point. Adjustments to the probe’s position are made along this normal direction, ensuring that the ultrasound probe maintains optimal alignment with the scanning surface. The new target position is determined as:$$\mathrm{Target Position}=\mathrm{Surface Point Position}+\mathrm{Target Surface Normal}\cdot d_{\mathrm{adjusted}}$$

#### 2.3.2. Path Planning

To achieve a higher level of automation, we equipped our robot with autonomous scanning path planning, implementing two distinct methods: direct path planning and area coverage path planning.

We first designed a user interaction interface to enable users to select path points and save them in sequence. The interface, shown in Figure 5, includes a 3D model of the patient’s abdomen and a simulated robotic arm, facilitating intuitive interaction. Using the Visualization Toolkit 7.1 (VTK) in the C++ library, we created an interactive window that visualizes the path point selection process.

In direct path planning, users manually select specific points on the 3D model by right-clicking on the desired locations. For each selected point, the system retrieves its 3D pose as a 4 × 4 transformation matrix, which encodes both the position and orientation information. These poses are stored sequentially to define a predefined scanning path. A red sphere visually marks the selected points on the 3D model, providing immediate feedback to the user, as shown in Figure 5a. This method is particularly suitable for small scanning areas with clear anatomical landmarks, such as the belly button or nipples, allowing for precise path customization.

In area coverage path planning, users define a rectangular scanning region by clicking and dragging the mouse over the desired area on the model. This action prompts VTK to capture the start and end points of the drag, which are then used to compute the four corner coordinates of the rectangle (Figure 5b). Based on these coordinates and the size of the ultrasound probe, a serpentine (zig-zag) path is generated to ensure complete coverage of the selected region. These path points are automatically calculated and stored in sequence for the robot to follow during scanning. This method is particularly advantageous for larger regions without clear anatomical landmarks, offering a more automated solution.

By integrating path planning with force control, the robot dynamically maintains safe and accurate alignment with the surface during scanning, ensuring optimal image quality and patient safety.

### 2.4. Tissue-Mimicking Phantom

To validate the force control mechanism of the robot, an abdominal phantom was developed. This model is critical for assessing abdominal aortic aneurysm (AAA), combining an idealized AAA model and an inferior vena cava (IVC) based on published vessel measurements from computed tomography and ultrasound scans. The abdomen phantom includes a stiffer layer containing the AAA and IVC, along with an outer adipose tissue layer to interact with the robotically controlled ultrasound (US) probe (Figure 6a).

Three-dimensional models of the lumens of the AAA and IVC were created based on aggregate dimensions from CT scan measurements. Models were 3D-printed using a desktop fused filamented fabrication printer. The AAA and its branches were printed in polyvinyl alcohol (PVA), a water-soluble synthetic resin, while the box and the unbranched IVC were printed in polylactic acid (PLA). Both pieces were coated with epoxy resin to create a smooth surface finish.

To create the stiffer inner layer of the phantom, a polyurethane rubber, Vytaflex 20 (Smooth-On Inc., Macungie, PA, USA), was chosen due to its Shore hardness of 20A and elastic modulus of E=337kPa. The urethane rubber was degassed in a vacuum chamber to eliminate air bubbles—a process essential to minimizing artefacts in imaging. The mixture was then poured into the box until the AAA and IVC lumen models were just covered and left to cure.

Following curing, the PVA model was dissolved in a heated and stirred water bath. The PLA model was removed by applying force to one end of the model.

To make the phantom more realistic to human abdomen tissue, the outer adipose layer simulating subcutaneous fat was added. A mixture of BJB F116 and BJB SC22 polyurethane rubber (BJB Enterprises, Tustin, CA, USA) was used, resulting in a flexible, realistic layer with a shore hardness of 00-15.

An ultrasound image of the phantom is shown in Figure 6b. For optimal imaging, ultrasound gel was applied between the two layers, and the phantom was submerged in water to a level above the vessels and below the probe. Scanning was performed using an EPIQ 7G Philips ultrasound machine (Bothell, WA, USA) and the Philips X6-1 ultrasound probe, optimized for abdominal imaging with a frequency range of 6–1 MHz. Minimal force was applied to the probe, which was sufficient to clearly delineate the boundaries of the two embedded vessels, mimicking realistic visualization conditions.

The ultrasound image reveals distinct vessel boundaries with minimal signal artefacts, indicating an effective simulation of abdominal structures. The contrast and clarity in the image suggest that the phantom’s materials and layer composition are appropriate for testing the force control and imaging capabilities of the robot, especially for applications like AAA assessment. This setup provides a reliable platform for evaluating the interaction between the robotically controlled ultrasound probe and simulated soft tissue, enhancing the realism and utility of the testing environment.

### 2.5. Experimental Design and Setup

This section outlines the design and setup of experiments conducted to validate the system’s performance, focusing on sensor accuracy and robotic control. The force sensor testing experiments ensured the reliability of the custom-designed sensor when integrated with the robotic system. The force-control algorithm testing experiments were designed to validate the robot’s ability to maintain target forces during sweeping motions, a key factor for ensuring patient safety. Finally, the automated ultrasound image acquisition experiments on phantom and volunteer subjects were designed to validate the system’s ability to acquire clear, clinically usable images under real-world conditions, such as patient breathing and varying tissue properties.

#### 2.5.1. Experimental Setting for Force Sensor Calibration and Performance Evaluation

As mentioned in Section 2.2.1, four load cells are evenly spaced on the same level and positioned on the lower part of the force sensor. As the four load cells share the total Z-axis force, the final sensor reading is obtained by summing the individual readings from each load cell. The force sensor calibration was conducted by applying known weights ranging from 0 g to 3500 g in increments of 500 g (Figure 7a). For each weight, the corresponding sensor readings were recorded. The calibration function was determined by calculating the linear fit of the sensor’s raw data against the known weights. After deriving the calibration function, the sensor’s accuracy, stability, and hysteresis were evaluated through further testing. To assess stability, known weights were applied to the sensor and held for 5 s, during which all sensor readings were recorded. The standard deviation of these readings was calculated to quantify the sensor’s stability. Accuracy was evaluated by averaging the sensor’s readings over the same period and comparing the result with the applied weight. Finally, hysteresis testing involved sequentially applying and removing weights one by one over five cycles. The sensor readings during the loading and unloading phases were compared to identify any discrepancies or lag in the sensor’s response.

We designed another experiment to test the sensor performance under the real ultrasound scanning scenario. In this experiment, the sensor was mounted onto the robot’s end effector to assess its performance under real ultrasound scanning conditions. In this experiment, the force applied by the robot arm was unknown, so an ATI Mini40 force sensor (ATI Industrial Automation, Apex, NC, USA) was used as a reference. As shown in Figure 7b, the ATI sensor was securely mounted on a custom-designed support platform to ensure stability and prevent slippage. The robot arm was positioned at various angles along different axes, including both in-plane and out-of-plane rotations, to push the ATI sensor and mimic real ultrasound scanning scenarios. A total of 32 poses were tested to evaluate the performance of our custom-designed force sensor. These poses included straight pushes along the Z-axis, lateral angles ranging from −25° to +25° in increments of 5°, elevational angles from −25° to +25° in 5° increments, and axial rotations from −60° to +60° in increments of 20°. Because the ATI sensor measures force along the global Z-axis, while our custom sensor measures force relative to the ultrasound probe’s axis, we calculated the projected force component on the global Z-axis for comparison with the ATI readings. It is important to note that the ATI sensor’s readings include the weight of the probe and the sensor itself, in addition to the force applied by the robot arm. These weights were premeasured and subtracted from the ATI’s readings.

#### 2.5.2. Experimental Setup for Force-Control Algorithm

In order to verify the usability and reliability of the surface force-following mode in the Z direction, we designed a platform for conducting the standalone force control experiment, as shown in Figure 8. The whole experiment was divided into the following steps: (1). One AprilTag marker and an ATI sensor were securely mounted on a 3D-printed platform, with the sensor fixed using screws. Another AprilTag marker was attached to the robot arm. (2). A depth camera was used to collect the pose data of the AprilTag markers on the platform and on the wrist of the robot in real time. (3). The platform was moved up and down by hand and we observed whether the robot follows the movement of the platform. In this process, we set the desired force range to be 3–6 N. (4). We observed the sensor data, i.e., the change in contact force, during the control of the platform moving up and down, and tried to keep the force values at less than 3 N, 3 N–6 N, and more than 6N, each for a certain period of time.

To assess the surface force-following function with the path-planning algorithm, we performed an automated scan on our AAA phantom (without the adipose layer). Initially, the 3D surface of the phantom was scanned and imported into the software. Users can employ the two-point rectangle method to manually select the upper-left and lower-right corners of the rectangle, after which the software automatically calculates the remaining two corners to define the full scanning area, as shown in Figure 9. The robot was programmed to execute a full-coverage scan within this designated area, aiming to maintain a consistent contact force ranging between 3 N and 6 N throughout the process. This force range was manually set to ensure optimal contact between the probe and the phantom surface. However, since the ideal force range varies among individuals, this range will be adjusted accordingly for volunteer scans.

#### 2.5.3. Experimental Setup for Automated Ultrasound Image Acquisition

Following the successful validation of the integrated functionality of the path-planning-based force-control algorithm, the system was deemed ready for automated ultrasound image acquisition using a real ultrasound probe. The setup for this acquisition process is illustrated in Figure 10. To mimic clinical imaging conditions and minimize surface friction, the AAA phantom was submerged in water, which eliminated air bubbles that could interfere with ultrasound signal transmission. The water level was carefully maintained below the ultrasound probe to ensure smooth probe movement without submerging the probe. Ultrasound gel was also applied between the probe and the phantom to ensure optimal contact. The custom-designed force sensor securely housed the ultrasound probe and was mounted on the robot’s end effector. Two AprilTag markers and one external camera were used to register the phantom with the robot. A standby operator was standing near the robot and phantom to monitor the entire process, ready to press the emergency stop button if necessary. For added safety, this emergency stop button will be given to volunteers or patients in human trials.

The phantom test procedure follows the same methodology as in Section 2.5.2, where the robot autonomously plans its path to cover the user-defined region while maintaining applied force within a specified range. In this experiment, however, our primary focus shifted to evaluating the system’s ability to acquire ultrasound images and analyzing the image quality obtained under varying force levels.

Lastly, we conducted volunteer tests to evaluate the system’s performance under real-world conditions, including patient breathing and variations in tissue stiffness. We obtained ethical approval from the King’s College London local ethics committee to test our robotic ultrasound system on volunteers for general abdominal scans (study title: Investigating Robotic Abdominal Ultrasound Imaging, study reference: HR-22/23-5412). Although our long-term goal is to develop a universal automatic ultrasound scanning protocol, this study first focused on the imaging of the abdominal aorta and common iliac arteries as an initial application due to the relatively straightforward motion planning requirements.

Two volunteers with different body shapes participated in the study: Volunteer 1 with a body mass index (BMI) of 21 and Volunteer 2 with a BMI of 33. The experimental setup, illustrated in Figure 11, was similar to the phantom test. In this configuration, the volunteer lay on a bed, with the RGB-D camera and robot positioned on their left side. The robot held an ultrasound probe connected to the ultrasound machine, which was located on the volunteer’s right side. The ultrasound system used was the Philips EPIQ 7G paired with the X6-1 probe (Philips Healthcare, Amsterdam, Netherlands), consistent with the phantom test setup. For safety, each volunteer was provided with an emergency stop button to hold during the procedure. If pressed due to discomfort, the robot’s power was immediately cut off, halting all operations to ensure the volunteer’s safety and comfort.

The volunteer test consisted of three distinct stages: (1) a manual scan performed by the operator, (2) a robotic scan conducted at the same position with varying levels of applied force, and (3) an automated robotic sweep designed to cover the entire region of interest.

Unlike scanning a phantom, which contains only two vessels and is composed of materials specifically designed for ultrasound imaging, the anatomy of the human abdomen is far more complex and varies between individuals. Therefore, the initial step is to locate the abdominal aorta or the common iliac arteries for each volunteer. To achieve this, we performed a freehand ultrasound scan. The scan is conducted exclusively in the transverse plane (where the aorta appears as a circular structure in the ultrasound image), as the diameter of the aorta is a critical criterion for detecting aneurysms. We initially positioned the probe on the volunteer’s abdomen, several centimetres to the left of the umbilicus. In the resulting ultrasound image, the top corresponds to the volunteer’s skin surface, while the bottom represents the spine and posterior structures. There are a number of key features used to identify the aorta. It appears as a circular structure with bright walls and a dark lumen. This is distinguished from the inferior vena cava (IVC), which has thinner walls and therefore appears less bright. The IVC is also less circular than the aorta due to the thinner walls and reduced intraluminal pressure (Figure 12a). This distinction can be further confirmed by the pulsing motion observed in B mode in the aorta, which is particularly clear in slimmer individuals. The pulsatile nature of the aorta can also be demonstrated with colour and spectral Doppler (Figure 12b), which contrasts to the respiratory modulated phasic flow demonstrated in the IVC. Anatomically, the aorta is situated just superficially to the spine, and to the left of the IVC. The spine appears as a curved structure at the bottom of the image, with a bright boundary and dark interior. Throughout this process, the sonographer must also adjust the ultrasound machine settings, including gain, depth, and focus, to optimize the image quality. These parameters are patient-specific and are carefully fine-tuned to ensure clear visualization of the aorta and surrounding structures.

After identifying the location of the aorta or iliac artery, the next step was to determine the optimal force range required to acquire high-quality ultrasound images for each volunteer. This force range was later used during the automatic scanning process. In this step, the ultrasound probe was mounted on the robot, and the system was operated in “kinematics control” mode to vary the probe’s depth in the Z-direction. Force values and their corresponding ultrasound images were recorded across a range of forces, from 2 N (representing minimal full contact) to 10 N. The results are presented in Figure 13. For Volunteer 1 with BMI = 21, optimal image quality was observed within the force range of 3–6 N. For Volunteer 2 with BMI = 33, optimal images appeared at 4–10 N. However, to balance image quality with patient comfort, the desired force range for automatic scanning was set to 4–7 N. As the applied force increases, depth to the centre of the aorta/iliac artery decreases. This occurs because the increased force causes the abdominal tissue to deform downward, compressing and displacing any gas pockets, which brings the probe closer to the aorta/iliac artery.

The third step mirrored the process used in the phantom test: selecting a region of interest within the robot control software and allowing the robot to automatically scan that area while maintaining the applied force within the predetermined range. For both volunteers, the region around the umbilicus was selected, with efforts to position the manual scan point at the centre of the targeted area.

## 3. Experiments and Results

This section mirrors the structure outlined in Section 2.5, presenting the corresponding results of the experiments described earlier. It systematically reports the outcomes of force sensor accuracy testing, the performance of the force-control and path-planning algorithms, and the system’s capability to acquire ultrasound images on both a phantom and a healthy volunteer.

### 3.1. Force Sensor Testing Experiments and Results

#### 3.1.1. Sensor Calibration and Sensor Performance Using Known Weights

Table 1 provides the results of the force sensor’s calibration and performance evaluation. The Measured Weight column lists the force values recorded by the sensor when known weights were applied, showing close agreement with the ground truth across all tested weights. The Avg. Error (N) and Avg. Error (g) columns show the average deviation between the measured weight and the ground truth, which ranges from +0.015 N (1.5 g) at 4.907 N to +0.18 N (18.3 g) at 0 N, with no noticeable trend as the applied weight increases. Accordingly, the Percentage Error column expresses the average error as a percentage of the ground truth, ranging from 0.1% at 34.349 N to 0.69% at 19.628 N, showing relatively consistent accuracy across the weights. The STD (N) column represents the standard deviation of sensor readings over 5 s, which increases with the applied weight, from 0.006 N at 0 N to 0.17 N at 19.628 N, reflecting a slight decrease in stability with higher forces. Overall, the results demonstrate the sensor’s reliable accuracy and stability across the tested weight range, with only minor variations.

As shown in Figure 14a, the sensor exhibited excellent linearity, achieving an R^2^ value of 0.9994. Additionally, the hysteresis behaviour, illustrated in Figure 14b, was minimal, with the hysteresis loops nearly overlapping across all five testing cycles. Figure 15 provides separate hysteresis maps for the five testing cycles, offering a detailed view of the sensor’s performance in each cycle. The maximum observed deviation was 0.19 N during the third cycle, which corresponds to only 0.52% of the full-scale output. This small deviation demonstrates the sensor’s high precision, repeatability, and consistency across multiple testing cycles.

#### 3.1.2. Sensor Performance on Robot Under Real-World Scenario

The root mean square error (RMSE) between our sensor readings and those from the ATI sensor was 0.71 N, with a standard deviation of 0.70 N. Some error may be attributed to slight deformation of the robotic arm under higher force levels, which can alter the effective pushing angle from the preset values. Despite this potential source of variation, a strong linear correlation was observed between the two sensors, as shown in Figure 16, with an R^2^ value of 0.97. This high correlation indicates that our sensor can reliably measure the force exerted by the ultrasound probe on the patient across a range of pressures and poses. The error observed remains within the tolerance required for safe and effective abdominal scanning.

### 3.2. Force-Control Testing and Experiments

#### 3.2.1. Standalone Experiment on Force Control

To evaluate the sensor performance in real force-control scenarios, we cross-validated the ATI sensor readings on the platform with those of the force sensor integrated into the ultrasound probe, as shown in Figure 17. The comparison reveals a good agreement between the two sensors, particularly when contact forces are low; both sensors produce nearly identical readings under these conditions. As the contact force exceeds 8 N, a slight discrepancy emerges, with the ATI sensor registering slightly higher values than the custom force sensor. This minor difference is likely due to the custom sensor measuring force along the *Z*-axis of the ultrasound probe. At higher forces, any tilt in the platform may introduce a misalignment between the probe’s axis and the true force direction, resulting in a slight underestimation by the custom sensor. This alignment under complex conditions suggests the robustness of our force sensor in measuring contact forces accurately.

Next, we plotted the Z-coordinate trajectories of both the robot arm and the platform in the camera coordinate system, as shown in Figure 18. Throughout the process, the gap between the two positions consistently remained at 80 mm (±10 mm), which corresponds to the vertical distance between the two markers when the robot arm and the ATI sensor are in proper contact. This consistency demonstrates the robot arm’s ability to move upward under high force and downward under low force, effectively tracking the platform’s vertical movement. However, around the 65-second mark, a rapid decrease in the platform’s Z-coordinate led to a significant delay in the robot’s response, with the gap exceeding 90 mm. This discrepancy is attributed to the fixed proportional gain in the hybrid force–position control, which limits the robot’s ability to adjust its response rate to rapid changes in force. Consequently, when the platform descends quickly, the robot is unable to immediately follow. Additionally, occasional zero-force readings were observed due to electromagnetic interference from nearby motors. However, the control software is programmed to ignore these sporadic readings, ensuring they do not affect the robot’s performance.

#### 3.2.2. Combined Path-Planning and Force-Control Experiment on Phantom

For the combined path-planning and force-control experiment, the recorded force sensor data, as illustrated in the upper graph of Figure 19, reveal the variations in the probe–surface contact force during the area coverage scan. During the scan, the force-control system dynamically adjusts the probe’s contact force in real-time as it follows the designated path, operating continuously alongside path planning. As the robot approaches each new point on the path, the system compensates for variations in distance, trying to maintain the force within the target range. Throughout the process, the force fluctuates between 0 and 12 N, generally increasing when below 3 N and decreasing when above 6 N. The standard deviation of the force values is 1.9 N.

Analysis of the force sensor data distribution, as shown in the lower two graphs of Figure 19, reveals two prominent peaks around 3 N and 6 N, indicating that these force values are commonly maintained during the scanning process. Outside the optimal 3–6 N range, the distribution density decreases, suggesting that the system spends less time at force values beyond this interval. This pattern highlights the force control system’s stability within the target range. Additionally, the pie chart shows the distribution proportions, with a 49.3% probability of the force being within the optimal 3–6 N range and a 93.7% probability of it remaining within the broader 1–8 N range. However, there are instances where the force spikes to higher values, such as around 21 s and 40 s. This occurs because the material was too rigid, making precise force control more challenging. In Section 3.3.1 and Section 3.3.2, subsequent tests conducted on softer tissues, including the phantom with an added adipose layer and healthy volunteers, demonstrated smoother and more stable force readings.

To validate the path planning, we first scanned the phantom using RGB-D data, obtaining the abdominal phantom surface in the camera coordinate system. By tracking the AprilTag marker on the robot arm and applying the transformation matrix from the arm to the probe’s face centre, we monitored the probe centre’s movement in real time within the camera coordinate system. Overlaying the phantom surface and probe trajectory data on the same graph provided a clear visualization of the ultrasound probe’s movement across the abdominal phantom, enabling a detailed assessment of path accuracy and coverage, as shown in Figure 20.

The probe follows the anticipated trajectory, moving back and forth between the upper and lower boundaries of the rectangular scan area to achieve full coverage. Although only the centre point of the probe’s trajectory is plotted, the probe’s width, exceeding 15 mm, ensures complete area coverage in practice. The probe paths were not perfectly parallel because slight tilting occurred due to friction between the probe and the phantom surface during movement, as well as adjustments made in response to real-time force readings.

### 3.3. Automated Ultrasound Image Acquisition

#### 3.3.1. Phantom Tests

The image acquisition test followed a similar protocol to the standalone force-control experiments, so we used the same data analysis method to evaluate the force during the scan, as shown in the upper graph of Figure 21. In this experiment, 57.0% of the applied forces fall within the 3–6 N range, while 98% of forces remain within a ±2 N range (1–8 N). The standard deviation of the force readings is 1.59 N, which is lower compared to readings without the fat layer and ultrasound gel.

Figure 22 displays the ultrasound images acquired by the robot during the automatic sweep under varying force levels. Overall, the robot successfully generated clear abdominal aorta ultrasound images without human intervention. In these images, vessel walls appear as bright circular shapes, with water inside the circle and the surrounding area comprised of the phantom material (polyurethane). The smaller vein consistently appears clearer than the larger aorta, likely because it is closer to the phantom surface and, therefore, undergoes less attenuation.

We recorded the applied force for each ultrasound image and observed that the clearest images were obtained at force levels of 5 N and 6 N, while forces outside this range resulted in reduced image quality. The decline in quality at higher forces may be due to deformation of the phantom, impairing contact between the probe and the surface or compressing the embedded vessels—an issue unlikely to occur in real participants. In human subjects, signal strength and image clarity typically improve with increased force as the distance to the area of interest decreases and attenuation is reduced. However, excessive force can introduce artefacts and discomfort. To better understand this force–image relationship, we propose measuring the distance to the centre of the abdominal aorta (derived from the ultrasound depth scale) as a metric correlated with probe force and image quality in future human trials. These preliminary phantom tests demonstrate the robot’s ability to autonomously acquire high-quality abdominal aorta images, laying a foundation for further studies to optimize the balance between image quality, patient comfort, and operator considerations in clinical settings.

#### 3.3.2. Volunteer Tests

The system successfully captured clear images of the abdominal aorta for both volunteers with different body shapes, which were reviewed by a registered clinical vascular scientist (HCPC CS20525) with 10 years of vascular ultrasound experience. Example images are presented in Figure 23.

Figure 24 illustrates the force readings and distributions recorded during automated ultrasound scans on two volunteers. The duration of each scan varies depending on the size of the selected region. The top graphs for both volunteers show a characteristic oscillatory pattern because of breathing motion, which was not observed in phantom tests. Regarding force distribution, Volunteer 1 demonstrated a lower proportion of forces within the desired range (39.6%) compared to Volunteer 2, where 65.1% of the forces fell within the target range. This difference can be explained by the higher BMI of Volunteer 2, where increased body fat and softer tissue facilitated smoother force control. The standard deviation of the forces further supports this observation, with Volunteer 1 exhibiting a higher standard deviation of 2.1 N compared to 1.6 N for Volunteer 2. The greater variability in Volunteer 1’s force readings is also attributed to the relatively firmer tissue, which made consistent force application more challenging. Nevertheless, the majority of force readings for both volunteers were within the ±2 N range, accounting for 93.1% and 96.7% of the forces for Volunteer 1 and Volunteer 2, respectively. The minor deviations outside this range primarily occurred at the beginning of the scan, as the robotic system transitioned from zero force to establishing surface contact. This is evident in the initial segments of the force value graphs. Throughout the scans, both volunteers reported no discomfort or pain.

## 4. Discussion

In this study, we developed a compact, single-axis ultrasound probe force sensor and implemented an automated ultrasound acquisition method for AAA using a custom-designed ultrasound robot. By combining path-planning with surface force-following control, the system enables full-coverage scanning within a defined area while maintaining a consistent force range, demonstrating promising potential for safe and effective robotic ultrasound scanning.

The custom-designed force sensor offers multiple advantages for robotic ultrasound applications. It is inexpensive and easy to manufacture with 3D printing, making it accessible for research and clinical use worldwide. Additionally, the sensor’s round shape and probe-holding mechanism mimic the grip of a human hand, ensuring stable contact. Sensor performance tests revealed high linearity and accuracy, with minimal hysteresis, indicating that the sensor is well suited for abdominal scanning. Experiments demonstrated that the force-control algorithm effectively responds to platform movements, maintaining a consistent distance between the markers on the robot’s wrist and the platform while dynamically adjusting the probe’s position to ensure optimal force application.

Integrated experiments combining force-following control with path planning validated the system’s ability to maintain a stable scanning trajectory while keeping contact force within safe thresholds, which is critical for patient comfort and safety. Additionally, it successfully acquired diagnostically usable abdominal aorta ultrasound images from two volunteers with different body shapes, demonstrating adaptability to variations in tissue properties and body composition. This adaptability marks a crucial step toward expanding the use of robotic ultrasound systems across diverse patient populations in healthcare settings.

Despite its advantages, the current sensor design has limitations. It measures force along only a single axis, limiting its ability to detect angular misalignments. To address this, future iterations should explore multi-axis force sensing, utilizing the four-load-cell design to enable three-axis measurements. This enhancement, coupled with an improved control algorithm, would allow for corrective tilting and more precise alignment with complex anatomical surfaces. Additionally, the current force-following control relies on a fixed proportional gain, which limits adaptability to sudden changes in surface position, such as those caused by patient movement or respiration.

In addition to addressing the current limitations, we are exploring the integration of image feedback into the system. This involves developing methods to automatically assess ultrasound image quality in real time and leveraging this feedback to optimize the probe’s position and orientation. Furthermore, diagnostic scans for AAA, as specified by the National Abdominal Aortic Aneurysm Screening Programme (NAAASP) [20] guidelines, must encompass the region from the diaphragm to the iliac bifurcation, necessitating precise probe alignment and positioning. To ensure accurate measurements, it is crucial to avoid oblique angles when the aorta is not parallel to the probe. These advancements represent key areas for future development aimed at enhancing the system’s diagnostic utility and clinical applicability.

## 5. Conclusions

In conclusion, our system offers a solution for safe human–robot interaction in extra-body ultrasound imaging acquisition. By employing multi-sensor fusion—including patient surface mapping, force sensing, ultrasound imaging, marker-based patient-to-robot registration, and scanning path planning—the system ensures both the safety and accuracy of sonographer-free, automated scanning. We hope that this work will contribute to the clinical acceptance of automatic and autonomous robotic ultrasound scanning systems, paving the way for broader adoption of such technologies in the future.

## Figures and Tables

**Figure 1 sensors-25-00468-f001:**
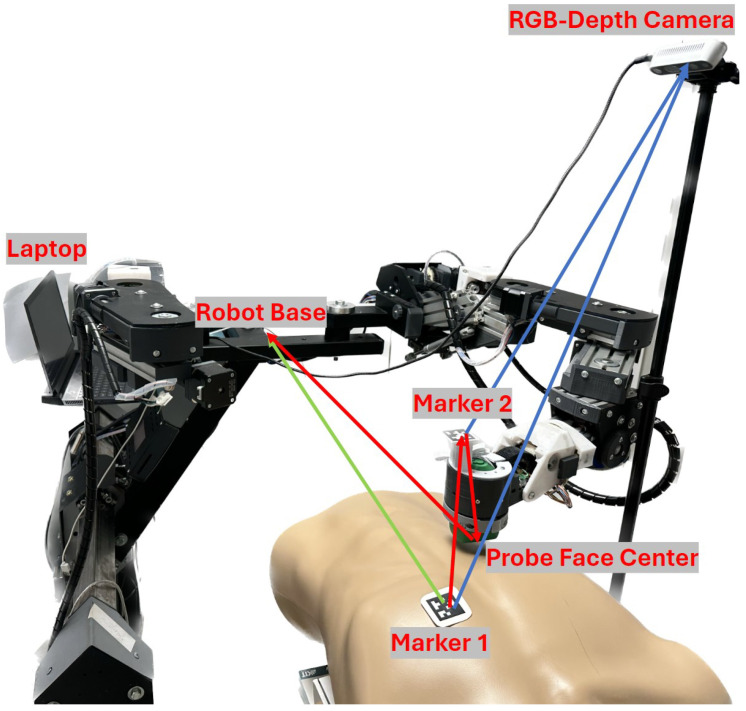
Patient-to-robot base registration process, along with three transformations utilized in this study shown with red arrows, relative marker transformation calculation with the help of the camera with blue arrows, and final patient-to-robot base transformation with the green arrow [19].

**Figure 2 sensors-25-00468-f002:**
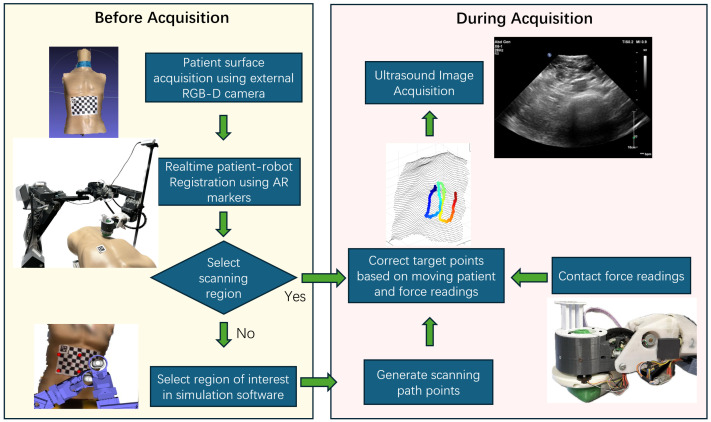
Workflow for automatic ultrasound image acquisition of the current system. The colorful path in the ’Correct Target Points’ section represents the corrected path plan, with blue indicating the start and red indicating the stop point. For detailed explanation, see Figure 20.

**Figure 3 sensors-25-00468-f003:**
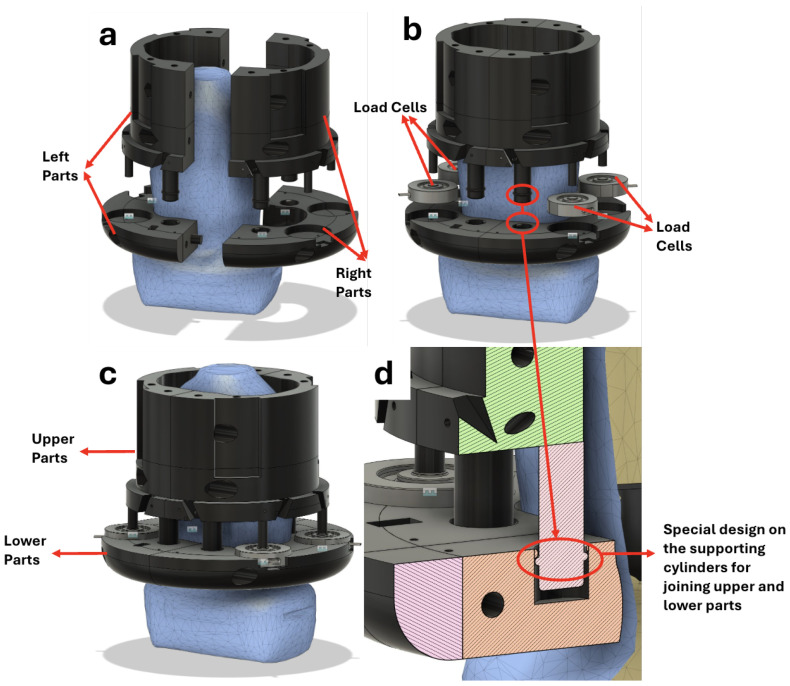
Custom-made force sensor mechanical structure design. (**a**) Left and right parts are separate to fit the ultrasound probe; (**b**) upper and lower parts are separate for easy replacement of the load cells; (**c**) complete sensor structure with probe and load cells; (**d**) special design on the supporting cylinders.

**Figure 4 sensors-25-00468-f004:**
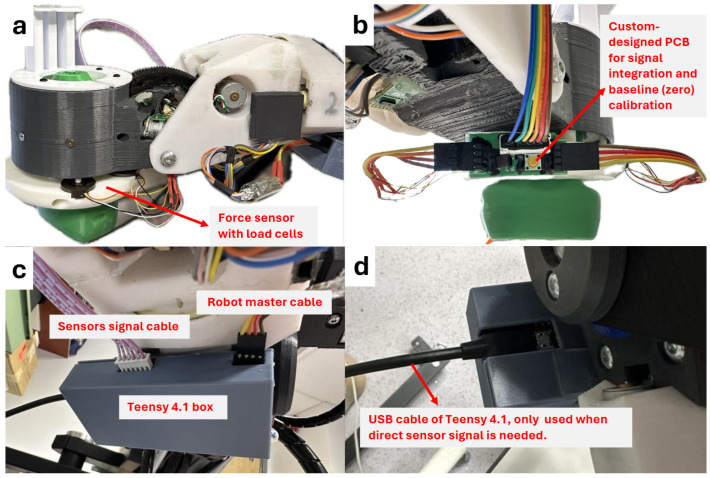
Force sensor signal processing setup. (**a**) Front view of the force sensor with integrated load cells. (**b**) Back view of the force sensor showing the custom-designed printed circuit board (PCB) for signal integration and zero calibration, along with two 4-pin input cables (each carrying signals from two load cells) and a 6-pin output cable connected to the Teensy 4.1. (**c**) Bottom view of the 3D-printed box for the Teensy 4.1 board, with sensor signal and robot master cables. (**d**) Top view of the Teensy 4.1 box, highlighting the USB cable used for direct sensor signal access when needed for debugging.

**Figure 5 sensors-25-00468-f005:**
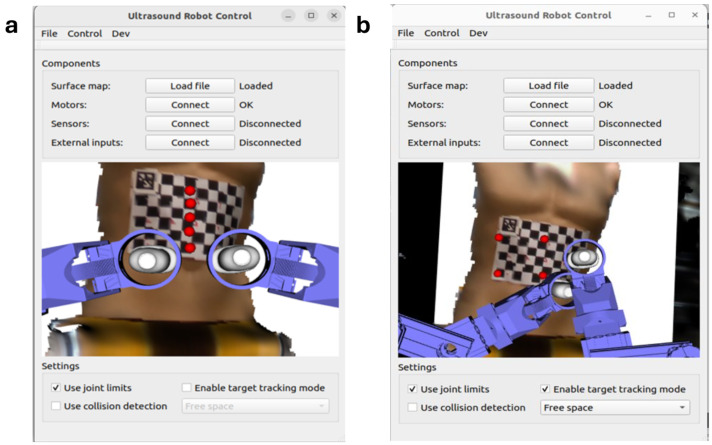
User interface for two path planning methods: (**a**) direct path planning method; (**b**) area coverage path planning method.

**Figure 6 sensors-25-00468-f006:**
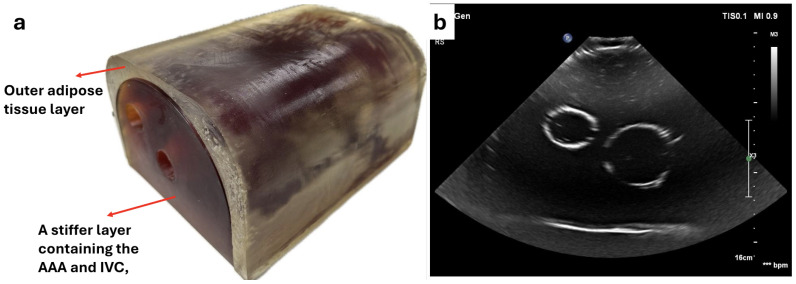
Ultrasound-compatible abdominal aortic aneurysm phantom: (**a**) external appearance and (**b**) ultrasound visualization of internal structures. The vertical dotted line in (**b**) represent a depth scale bar, with each segment indicating 1 cm.

**Figure 7 sensors-25-00468-f007:**
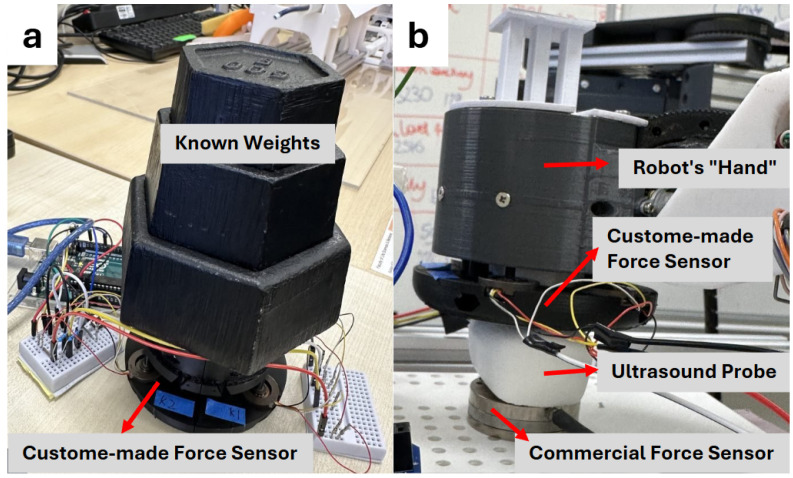
Sensor calibration and performance assessment: (**a**) on its own with known weights; (**b**) on the robot with an ultrasound probe and commercial force sensor.

**Figure 8 sensors-25-00468-f008:**
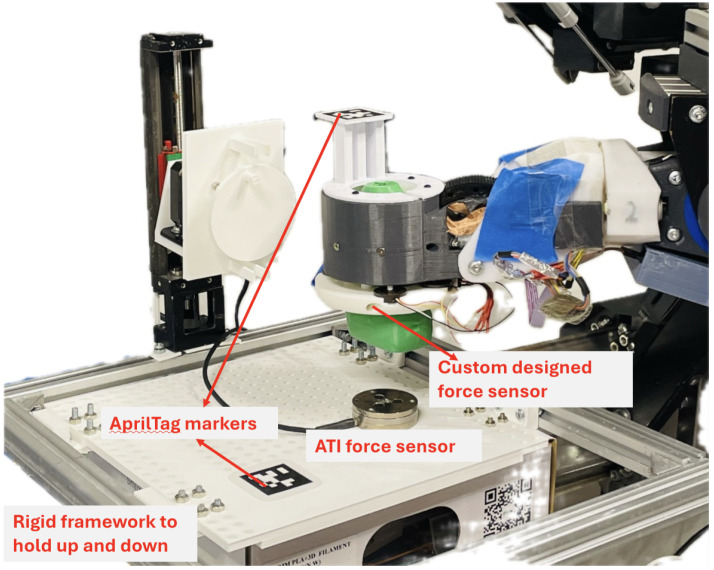
Experimental setup for validating the surface force-following algorithm in the Z-direction using an ATI force sensor, a 3D-printed framework, and two AprilTag markers used for position registration.

**Figure 9 sensors-25-00468-f009:**
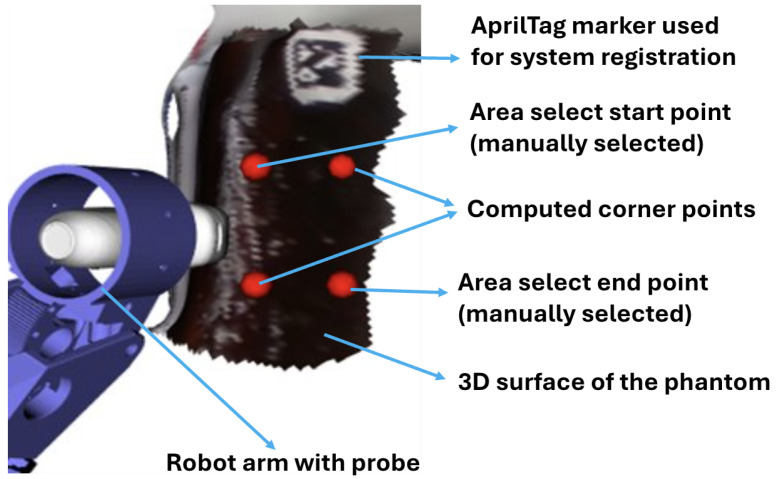
Selected area on abdominal phantom in simulation window.

**Figure 10 sensors-25-00468-f010:**
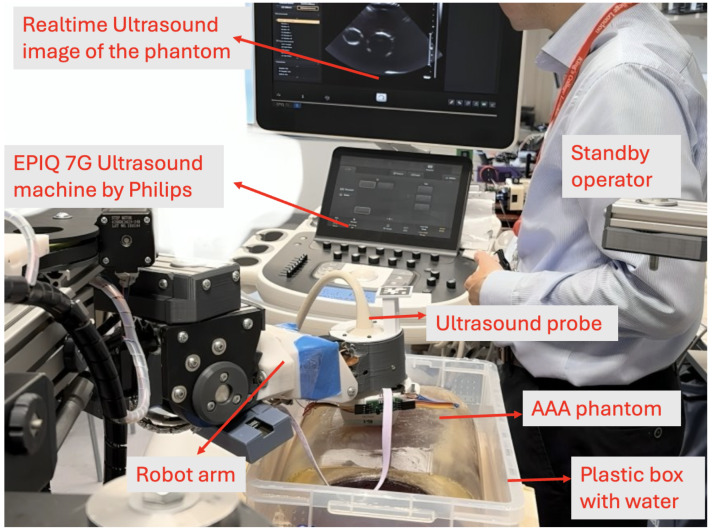
Automated ultrasound image acquisition on phantom experiment setup.

**Figure 11 sensors-25-00468-f011:**
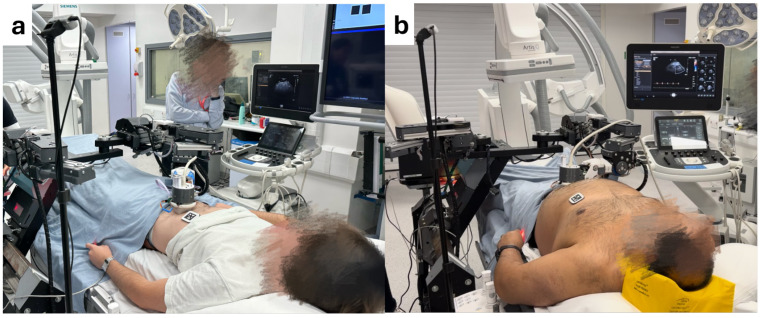
Setup for automated ultrasound image acquisition in healthy volunteer tests: (**a**) Volunteer 1 with BMI = 21, (**b**) Volunteer 2 with BMI = 33.

**Figure 12 sensors-25-00468-f012:**
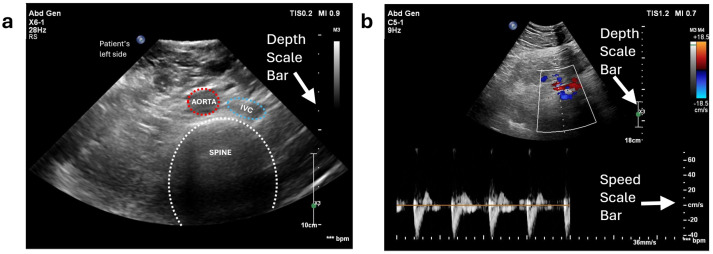
Abdominal aorta and iliac artery localization in volunteer testing: (**a**) B−mode image of the abdominal aorta annotated by a clinical vascular scientist (Volunteer 1) and (**b**) pulsed wave Doppler of the iliac artery (Volunteer 2).

**Figure 13 sensors-25-00468-f013:**
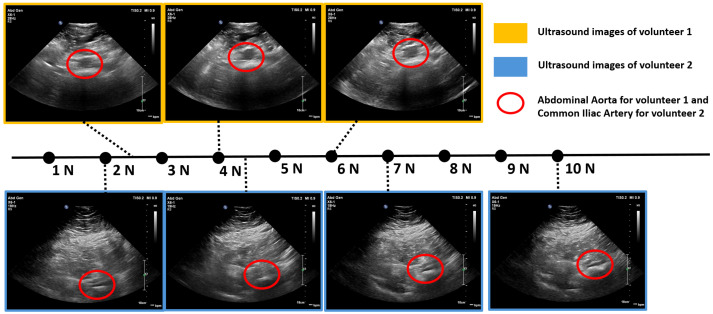
Ultrasound images acquired by the robot at different force levels: Volunteer 1’s abdominal aorta (yellow border) and Volunteer 2’s iliac artery (blue border). The vertical dotted line in all the ultrasound images represent a depth scale bar.

**Figure 14 sensors-25-00468-f014:**
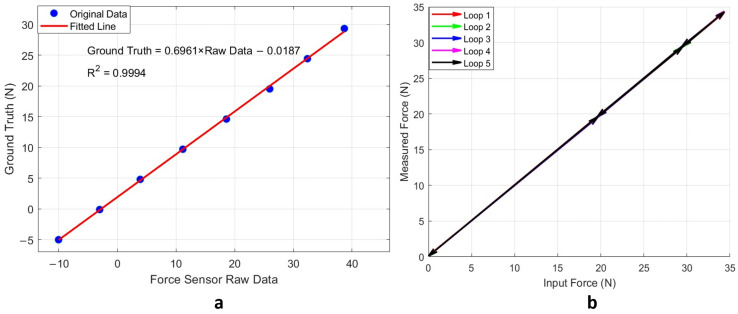
Performance testing of the force sensor using known weights: (**a**) linearity and calibration equation relating sensor raw data to ground truth and (**b**) hysteresis evaluation over five loading–unloading cycles, comparing input and measured weights. The five colored lines overlap, resulting in a single visible black line. Detailed zoomed-in graphs are provided in Figure 15.

**Figure 15 sensors-25-00468-f015:**
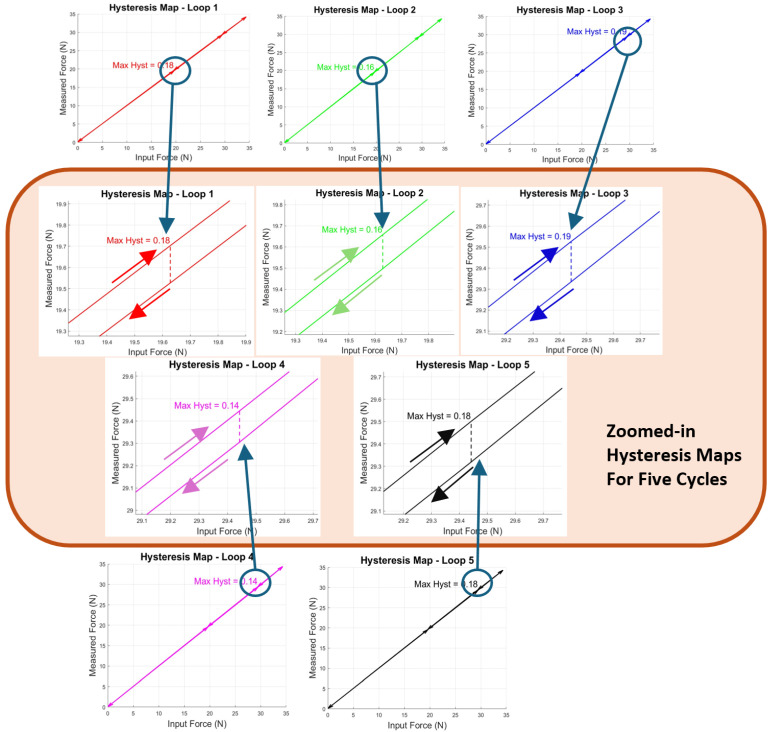
Separate hysteresis maps for five cycles with zoomed-in largest hysteresis points highlighted in the orange rectangle.

**Figure 16 sensors-25-00468-f016:**
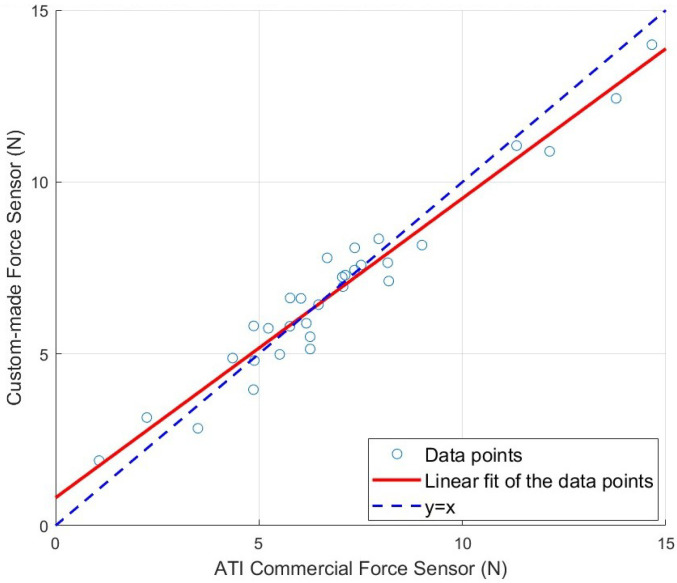
Correlation between the ATI commercial force sensor readings (*x*-axis) and our custom-made ultrasound force sensor readings (*y*-axis). Each blue point represents a measurement instance where the same force was applied to both sensors. The red solid line indicates the linear fit of these data points, while the blue dashed line represents the ideal 1:1 correspondence (y = x) between these two sensors.

**Figure 17 sensors-25-00468-f017:**
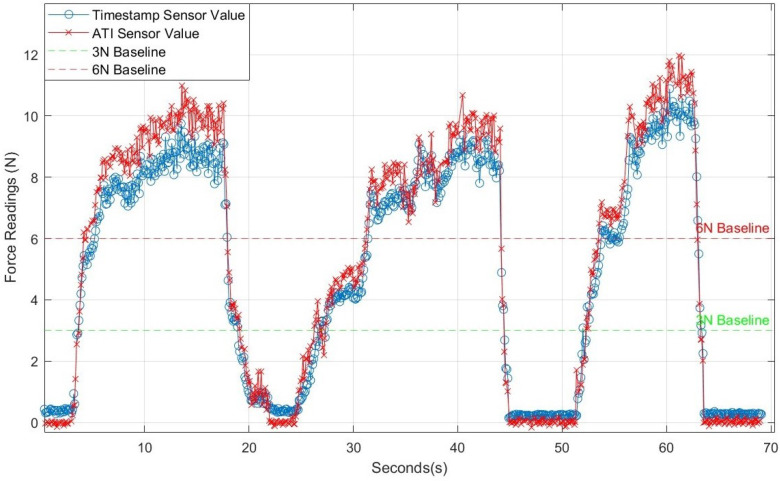
Force comparison between custom-made force sensor and ATI force sensor.

**Figure 18 sensors-25-00468-f018:**
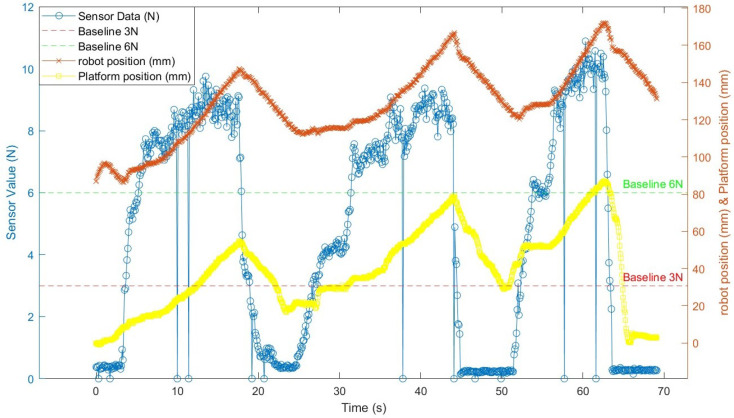
Force sensor data, platform position, and robot position over time.

**Figure 19 sensors-25-00468-f019:**
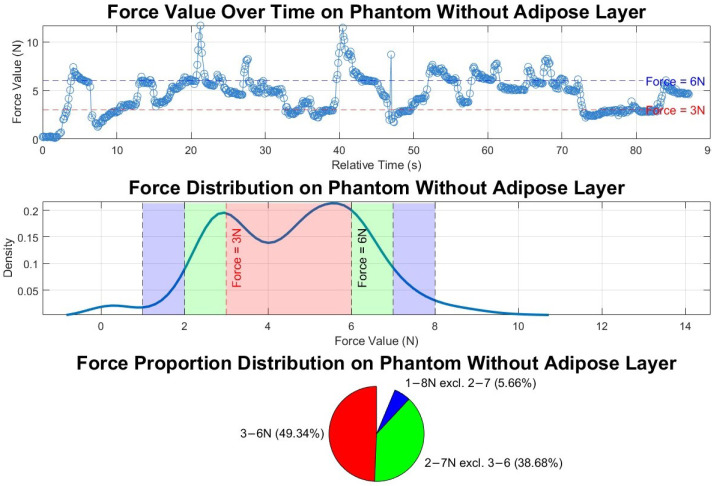
Force values and distributions in the combined path-planning and force-control experiment on phantom without adipose layer. The top subfigure shows force variation over time, with 3 N and 6 N thresholds marked; the middle subfigure depicts the force value distribution, highlighting ranges below 3 N, 3–6 N, and above 6 N; and the bottom subfigure displays the time proportion within each force range. Red, green and blue colours in the middle and buttom graphs match with each other.

**Figure 20 sensors-25-00468-f020:**
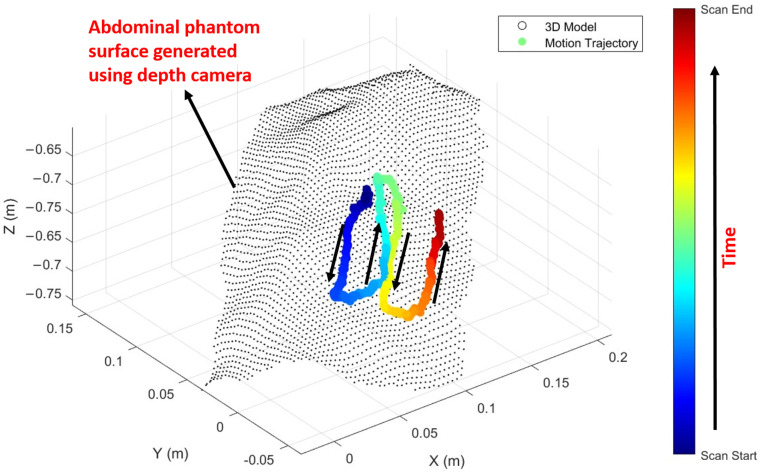
Probe’s 3D trajectory and the phantom surface visualization in the real-world coordinate system. The abdominal phantom surface is represented as a black grid and the probe centre’s 3D trajectory during the scanning process is colour-coded by time, transitioning from blue (start) to red (end).

**Figure 21 sensors-25-00468-f021:**
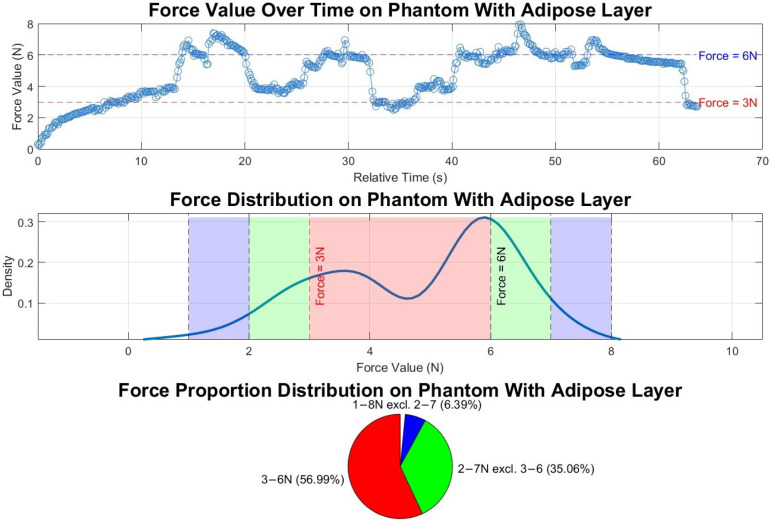
Force values and distributions in automated ultrasound image acquisition experiment on phantom with adipose layer. The top subfigure shows force variation over time, with 3 N and 6 N thresholds marked; the middle subfigure depicts the force value distribution, highlighting ranges below 3 N, 3–6 N, and above 6 N; and the bottom subfigure displays the time proportion within each force range. Red, green and blue colours in the middle and buttom graphs match with each other.

**Figure 22 sensors-25-00468-f022:**
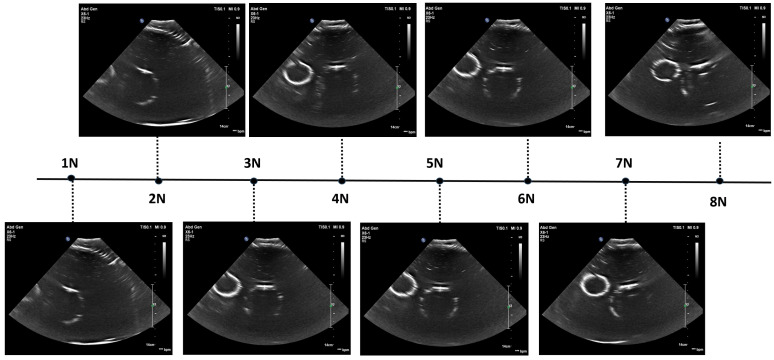
Ultrasound images acquired by robot at varying force levels on AAA phantom.

**Figure 23 sensors-25-00468-f023:**
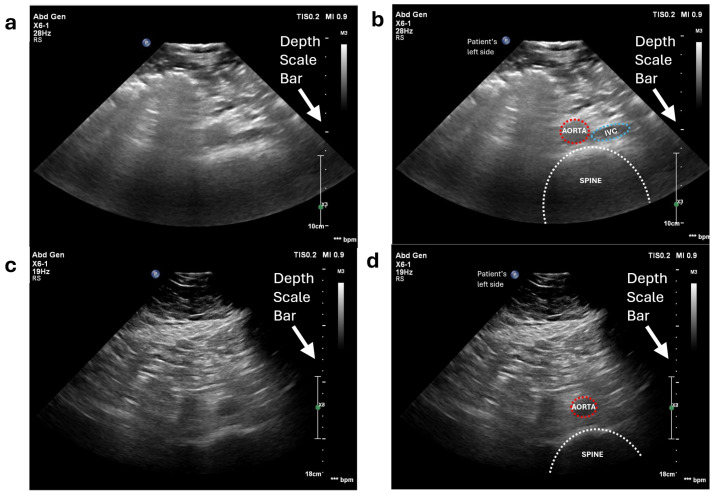
Abdominal aorta ultrasound images acquired by the robotic system: (**a**) Volunteer 1’s original image; (**b**) Volunteer 1’s images annotated by registered clinical vascular scientist; (**c**) Volunteer 2’s original image; (**d**) Volunteer 2’s images annotated by registered clinical vascular scientist.

**Figure 24 sensors-25-00468-f024:**
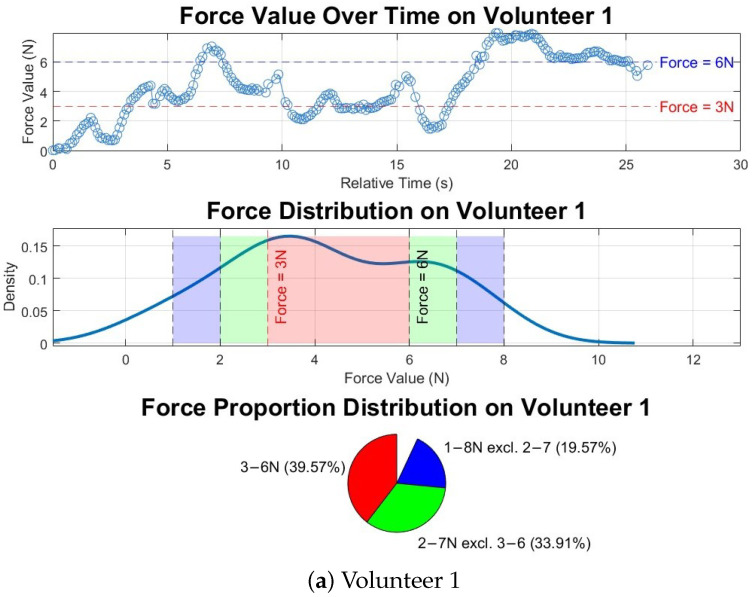
Force values and distributions in automated ultrasound image acquisition experiment on two volunteers. Red, green and blue colours in the middle and buttom graphs match with each other.

**Table 1 sensors-25-00468-t001:** Accuracy and stability test results using known weight.

Ground Truth	Measured Weight	Avg. Error (N)	Avg. Error (g)	Percentage Error	STD (N)
0 N	0.18 N	+0.18 N	18.3 g	0.31%	0.006 N
4.907 N	4.922 N	+0.015 N	1.5 g	0.31%	0.031 N
9.814 N	9.840 N	+0.026 N	2.7 g	0.26%	0.049 N
14.721 N	14.815 N	+0.094 N	9.6 g	0.64%	0.155 N
19.628 N	19.764 N	+0.136 N	13.7 g	0.69%	0.17 N
24.535 N	24.653 N	+0.118 N	12.0 g	0.48%	0.065 N
29.442 N	29.581 N	+0.139 N	14.2 g	0.47%	0.124 N
34.349 N	34.313 N	−0.036 N	3.7 g	0.1%	0.14 N

## Data Availability

Data are unavailable due to privacy or ethical restrictions.

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
