# Peer review of "Design of a Cost-Effective Ultrasound Force Sensor and Force Control System for Robotic Extra-Body Ultrasound Imaging"

_sensors, 2025, doi:10.3390/s25020468_

Round 1

Reviewer 1 Report

Comments and Suggestions for Authors

The paper is of high interest in proposing a robot-assisted ultrasound screening. The preliminary result is promising. However, Some amendments are necessary as listed below:

(1) 2.3.1 & 2.3.2 are too brief to understand. Please elaborate.

(2) Please provide the force/torque relations of the contact probe to the four load cells' outputs. How are the outputs from load cells used for control the probe pose so that the contact adjustment can be made for good quality of ultrasound imaging.

(3) Please provide reference for phantom design and fabrication. We don't know which part is novel, which part is based on previous work from the references.

Reviewer 2 Report

Comments and Suggestions for Authors

1. There is only one line can be seen in the Figure 14b, which is actually difficult to guarantee, and the author can zoom in to show the difference between different loops.

2. The article is relatively messy, what‘s the key point or the main contribution? the sensor or the ultrasonic robot system? And the two parts should be divided more clearly, and the sensor characterization should be concentrated together, first characterizing the sensor, and then testing the system performance.

3.The author conducted human experiments, whether this is ethical, 

4. The figures of the article should mainly be the characterization, control effect, and accuracy analysis of the sensor or system, which is not necessary for some software interfaces.

Round 2

Reviewer 1 Report

Comments and Suggestions for Authors

I still couldn't find any cited papers in references regarding the tissue-mimicking Phantom. 

Author Response

Comments 1: [ I still couldn't find any cited papers in references regarding the tissue-mimicking Phantom. ]

Response: Thank you for your comment. The abdominal phantom presented in this manuscript is a novel contribution developed entirely based on the expertise and know-how within our research team. It was not derived from or based on any previously published work. Therefore, no specific references were included regarding its development.